# Evolution of cooperation in costly institutions exhibits Red Queen and Black Queen dynamics in heterogeneous public goods

Mohammad Salahshour [1] ✉

Public goods are often subject to heterogeneous costs, such as the necessary costs to maintain the public goods infrastructure. However, the extent to which heterogeneity in participation cost can affect groups' ability to provide public goods is unclear. Here, by introducing a mathematical model, I show that when individuals face a costly institution and a free institution to perform a collective action task, the existence of a participation cost promotes cooperation in the costly institution. Despite paying for a participation cost, costly cooperators, who join the costly institution and cooperate, can outperform defectors who predominantly join a free institution. This promotes cooperation in the costly institution and can facilitate the evolution of cooperation in the free institution. For small profitability of the collective action, cooperation in a costly institution but not the free institution evolves. However, individuals are doomed to a winnerless red queen dynamics in which cooperators are unable to suppress defection. For large profitabilities, cooperation in both the costly and the free institution evolves. In this regime, cooperators with different game preferences complement each other to efficiently suppress defection in a black queen dynamic.

[1] Max Planck Institute for Mathematics in the Sciences, Inselstrasse 22, D-04103 Leipzig, Germany. ✉email: salahshour.mohammad@gmail.com

A collective action problem arises when individuals' and group's interests conflict. The provision of public goods is a prime example. From resource acquisition in bacterial populations[1,2], and cooperative defending, foraging[1,3,4], and breeding[5] in animal populations, to maintaining the commons such as fisheries[6], or environmental preservation[6,7] in human societies, the success of the groups relies on the contribution of individuals to provide public goods. While beneficial for all, the provision of public goods is costly for individuals. Selfish individuals who refrain from contributing to the public goods and free-ride on the contributions of others receive a higher payout. Indeed, a naive evolutionary argument suggests free-riding but not cooperation should prevail, resulting in a tragedy of the commons where everybody ends up worse off[8]. When individuals live in groups composed of highly related individuals, such as in eusocial insects, kin selection is argued to be able to avert a tragedy of the commons, based on the fact that the beneficiaries of a cooperative act is likely to be cooperators who share the cooperative trait[9]. However, groups are often composed of unrelated or weakly related individuals. By considering a context where individuals need to provide a public good, often modeled as a public goods game, past researches have shown that formal and informal social institutions play an important role in solving collective action problems in groups composed of unrelated individuals. Informal social institutions can be at work, for instance, in the form of social norms[10–17], reputation and gossip[10,18–20], or even human language[18,21]. Such institutions can promote cooperation by channeling the benefit of cooperation towards cooperators. Human institutions, formal or informal, can promote cooperation through other mechanisms as well, such as reward[22–26], and punishment[27–35]. Such institutions can help to solve collective action problems by rewarding social behavior or punishing anti-social behavior.

Most of the studies have so far considered a simplified context where individuals need to provide a public good. In contrast, in many realistic situations, individuals face different public good dilemmas at the same time. For instance, a growing body of experimental literature in the field of microbial social evolution has studied situations where microbes need to engage in different public goods dilemmas, and interesting phenomena, such as evolutionary arms race between different microbial strains engaging in a heterogeneous collective action problem is observed in recent years[36–39]. In the other extreme of biological complexity, humans in societies often face different public goods dilemmas, for example, in the form of different collective action institutions that they can join[40–44]. Recently, it has been shown that the interaction between public goods provides a simple mechanism for the solution of collective action problems in a context where multiple public goods exist[45]. However, public goods are heterogeneous in different aspects. Recently, the effect of different sorts of heterogeneity, such as heterogeneity in the network of interactions[46,47], resource availability[48,49], heterogeneity in public goods returns to individuals[50,51], and heterogeneity in the game structure[52] has been studied in public goods games. However, how heterogeneity affects interacting public goods is an interesting question that remains to be explored. A realistic constraint on public goods is that public goods may have heterogeneous costs associated with instituting and maintaining the public good. In the case of human populations, a cost can originate, for instance, from transaction costs associated with administrating the group and its infrastructure. In microbial populations, cost heterogeneity can originate from the cost associated with developing and keeping a public goods infrastructure, such as an iron-uptake system in siderophore-producing bacteria[36–39].

Here, I bring attention to a relatively simple effect that can promote cooperation in costly games when multiple public goods exist and interact. I show that in a situation where individuals have a choice between different institutions, the existence of an entrance cost to enter an institution can keep free-riders away. Free-riders engage in a risky enterprise by joining a costly institution: Free-riding in a costly institution can yield a positive outcome only when enough cooperators in the costly institution exist. Otherwise, by joining the costly institution, free-riders pay a pure cost. Consequently, as long as the expected number of cooperators in the costly institution is low, free-riders are better off joining a free institution to avoid potential losses. Thus, by increasing the likelihood of cooperation, an entrance cost functions as a precommitment for cooperation, and free-riders predominantly join the free institution. This fact makes a costly institution a cooperation niche, in which costly cooperators can work cooperatively and obtain a higher profit despite paying an entrance cost with no direct benefit.

As our analysis shows, in both well-mixed and structured populations, a costly institution outperforms a free institution when the collective action's profitability is small: While cooperation does not evolve in a free institution, it does evolve in a costly institution. However, the average payoff of the individuals from the game remains close to its entrance cost, and thus, their total payoff remains close to the same value that they would have obtained if the costly institution had not existed. In this regime, individuals engage in a winner-less red queen dynamics[53,54] in which the frequency of costly cooperators, costly defectors, and non-costly defectors shows cyclic fluctuations in a rock–paper–scissor manner.

As the profitability of the collective action increases, the system shows a transition to a phase where cooperation in both the costly and the free institutions evolve. In this regime, a synergistic mutualism between cooperators with different game preferences emerges. While vulnerable to defectors with the same game preference, cooperators can beat the defectors with a different game preference and thus help cooperators in their opposite institution. Consequently, a functional complementation between cooperators emerges, which gives rise to a black queen dynamics[36,55] in which cooperators with different game preferences complement each other to efficiently avoid exploitation by defectors.

On a spatial structure, the negative density dependence of different strategies can give rise to traveling waves, an evolutionary arms race in which defectors effectively chase cooperators, and cooperators, in turn, evolve to avoid being exploited by cheaters by mutating game preference. Depending on the profitability of the public goods and the entrance cost, the system can exhibit a winner-less red queen evolutionary dynamic in which competition between defectors helps the survival of costly cooperators or one in which cooperators with different game preferences complement each other to overcome the defectors and emerge as the winners.

As we argue, similar phenomenology is at work when instead of a costly and cost-free institution, two public goods institutions with a smaller and a higher cost interact. Our finding sheds light on the rich phenomena emerging from the competition between the individuals in interacting public goods and can help better understand the effect of heterogeneity in interacting public goods.

## The model

I consider a population of $N$ individuals. At each time step, groups of $g$ individuals are drawn at random from the population pool by decomposing the population into $N/g$ groups. Individuals in each group can choose between two public resources: A costly institution, which I call the resource 1, and a free institution, which I call resource 2. The costly institution has a participation

fee that is paid by all the individuals who choose this resource. Individuals gather payoff by playing a public goods game in their institution. In this game, individuals can either cooperate or defect. Cooperators pay a cost $c$ to invest the same amount in the public resource. Defectors pay no cost and do not invest. All the investments in a public resource $i$ are multiplied by an enhancement factor $r_i$ and are divided equally among the individuals in that resource. I assume individuals also receive a base payoff, $\pi_0$, from other activities unrelated to the public goods game.

After playing the games, individuals reproduce with a probability proportional to their payoff. In the reproduction stage, the whole population is updated such that the population size remains constant. That is, for each individual in the next generation, an individual is chosen as a parent with a probability proportional to its payoff. The offspring inherit the game preference (which can be institution 1 or 2) and the game strategy (which can be cooperation $C$ or defection $D$) of its parent, subject to mutations. Mutation in the game preference and strategy occurs independently, each with probability $v$. In case a mutation occurs, the value of the corresponding variable is changed to its opposite value (for example, 1 to 2 for the game preference and $C$ to $D$ for the strategy).

In addition to a well-mixed population, I consider a structured population, in which the individuals reside on a network. I consider a first nearest neighbor square lattice with von Neumann connectivity and periodic boundaries for the population network. Each individual participates in five groups, each centered around itself or one of its neighbors, to perform a collective action task. Individuals in each group enter and play a public goods game in their preferred institution. The payoff of the individuals from the public goods game is defined as their average payoff from all the games that the individuals participate in. In addition to the payoff from the public goods games, individuals receive a base payoff $\pi_0$. After deriving their payoffs, the whole population is updated. For reproduction, I consider an imitation or death-birth process in which each individual imitates the strategy of one of the individuals in its extended neighborhood, chosen with a probability proportional to its payoff. The extended neighborhood of an individual is defined as the individual itself together with its neighbors. I assume mutations can occur as well. After imitation, the individual's strategy and game preference mutate independently and each with probability $v$ (see Supplementary Note 1 for further details).

## Results

**Well-mixed population**. As shown in the Methods section, in a well-mixed population, the model can be described in terms of the replicator-mutator dynamics. I begin, by a case where the quality of the two resources are similar, $r_1 = r_2 = r$, and plot the frequency (solid blue), the average payoffs from the game (dashed red), and the amplitude of fluctuations (dotted blue) for different strategies in Fig. 1a–d. Here, the replicator dynamic is solved starting from a uniform initial condition in which all the strategies' initial frequency is equal. The results of simulations in finite populations are in good agreement with the replicator dynamics results (see Supplementary Note 2 and Supplementary Figs. 1, 2, 3, and 4 for comparison to simulations). Throughout this manuscript I fix $c = 1$.

For small enhancement factors $r$, the dynamics settle in a fixed point where only defectors in the free institution survive. The advantage of the costly institution becomes apparent as $r$ increases beyond $r^* = 1 + c_g$, such that the maximum possible payoff of the costly institution, which is achieved when nobody defects and is equal to $r - 1 - c_g$ becomes positive. As shown in the Methods, a focal defector in a group composed of $n_C^1$ costly

cooperators and $n_D^1$ other costly defectors (and $n_C^2 + n_D^2$ individuals who prefer the free resource), obtains a payoff of $n_C^1 r / (n_D^1 + n_C^1 + 1) - c_g$. This payoff becomes negative for a small enough value of $n_C^1$ (or a large enough value of $n_D^1$). Since groups with a small number of costly cooperators are drawn with a high probability when $\rho_C^1$ is small (these probabilities can be derived in terms of multinational coefficients, see the Methods), the average payoff of a costly defector remains negative in this regime (for instance, given at the transition $\rho_C^1 \approx \nu$, a mutant costly defector finds itself in a group with no costly cooperator with probability $(1 - \rho_C^1)^{g-1} \approx (1 - \nu)^{g-1}$, which is close to 1 for low mutation rates, and pays a pure cost of $-c_g$). On the other hand, a costly cooperator's payoff is equal to $(n_C^1 + 1)r / (n_D^1 + n_C^1 + 1) - c_g - 1$, which for small enough $n_D^1$ becomes positive. As in this region, the frequency of costly defectors, $\rho_D^1$, is small, such group compositions occur with a high probability (at the transition, $\rho_D^1 \approx \nu$, and thus the probability that a costly cooperator joins a group with no costly defectors is $(1 - \rho_D^1)^{g-1} \approx (1 - \nu)^{g-1}$, which is close to 1 for low mutation rates). Consequently, the average payoff of costly cooperators from the game becomes positive, and thus, larger than the dominant non-costly defectors' payoff (who receive a payoff of zero). Consequently, the frequency of costly cooperators rapidly increases at $r^*$. However, due to the rapid increase in the frequency of costly cooperators at $r^*$, the probability of formation of such mixed groups increases, and costly defectors start to appear in the system. Further increasing $r$ in this region, the frequency of costly cooperators and costly defectors increases at the expense of non-costly defectors.

As $r$ increases above a second threshold, cyclic fluctuations set in, and the dynamics settle in a periodic orbit. An example of this periodic orbit is presented in Fig. 2a, b. Interestingly, the average payoff of costly cooperators, costly defectors, and non-costly defectors in this region remains close to zero despite the evolution of cooperation. Although individuals constantly update their strategy to overcome others, no strategy wins in the evolution. Instead, individuals engage in a winnerless red queen dynamic. The game payoffs of costly cooperators and costly defectors fluctuate around zero (which is equal to the game payoff of non-costly defectors). The dynamics of the system in this regime resembles the frequency-dependent selection in the host-parasite evolution, coined the red queen dynamic based on the fact that no matter how much they run, all end up in the same place[53,54]. On this basis, I call this periodic orbit the red queen periodic orbit.

The existence of a costly institution can facilitate the evolution of cooperation in its competing free institution too. As the amplitude of fluctuations increases, episodes where most of the individuals prefer the costly institution occur. During these episodes, $\rho_D^2$ drops to a small value. Consequently, the probability that a mutant non-costly cooperator finds itself in a group devoid of non-costly defectors $(1 - \rho_D^2)^{g-1}$, increases. In such groups, non-costly cooperators receive a payoff of $r - 1$, which is larger than the payoff of all the other strategies and outcompete other strategies. At this point, a second periodic orbit emerges in which cooperation in both the costly and free institutions evolves. The evolution of cooperation in the free institution can, in turn, have a positive impact on cooperation in the costly institution. This is the case because above the point where cooperation in the free institution evolves, the frequency of individuals who prefer the free institution starts to increase by increasing $r$. This effect decreases the frequency of those who prefer the costly institution and its effective size. This decreases the mixing probability between costly cooperators and costly defectors and increases the costly cooperators' payoffs. Consequently, a functional complementation between cooperators with different game preferences

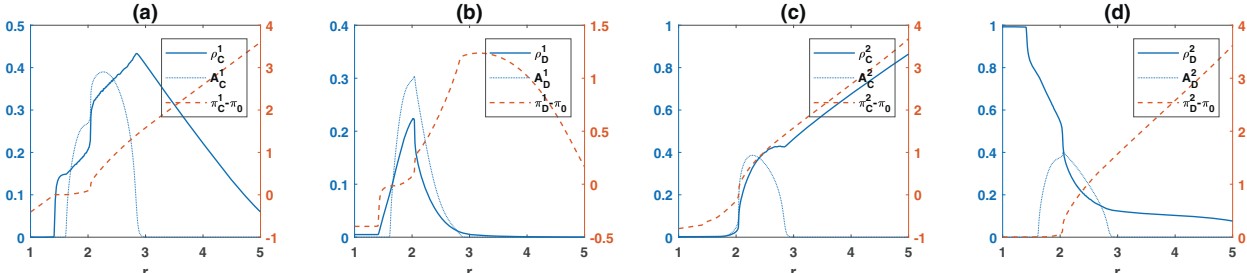

**Fig. 1 The frequency, game payoffs, and the amplitude of fluctuations for different strategies.** The frequency (solid blue), game payoffs (dashed red), and amplitude of fluctuations (dotted blue) of costly cooperators (**a**), costly defectors (**b**), non-costly cooperators (**c**), and non-costly defectors (**d**), as a function of the enhancement factor, $r$. As $r$ increases, above a first threshold ($r^* = 1 + c_g$), cooperation in the costly institution evolves, and above a second threshold (approximately $r = 2$) cooperation in both the costly and the free institutions evolves. For medium $r$, the system shows periodic fluctuations. Parameter values: $g = 5$, $nu = 10^{-3}$, $\pi_0 = 2$, $c_g = 0.398$. The replicator dynamic, derived in the Methods Section, is solved for 9000 time steps, and the time averages are taken over the last 2000 time steps.

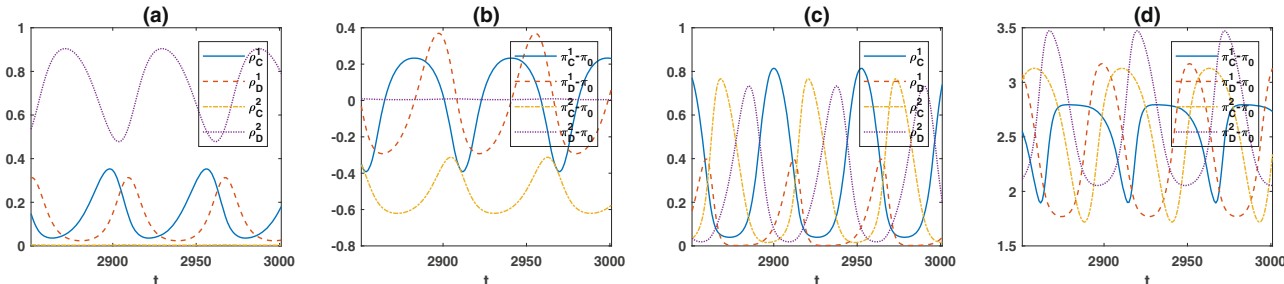

**Fig. 2 Red queen and black queen orbits.** The frequency of different strategies (**a**) and the game payoffs (**b**) in the red queen, and the black queen (**c**, **d**) periodic orbits. In the red queen orbit, cooperators in the costly institution survive. However, the payoff of the surviving strategies fluctuates around zero, and none dominate others. In contrast, cooperators in both institutions evolve in the black queen orbit, and cooperators of each type suppress defection in their opposite institution. Consequently, the payoff of all the strategies starts to deviate from zero. Parameter values: $g = 5$, $nu = 10^{-3}$, $\pi_0 = 2$, and $c_g = 0.398$. In (**a**, **b**) $r = 1.7$, and in (**c**, **d**) $r = 2.2$.

emerges, which is reminiscent of a black queen dynamics in which different types crucially depend on each other for performing vital functions[36,55]. While vulnerable to defectors in their own institution, cooperators complement each other by beating defectors in their opposite institution. Synergistically thus, they can suppress defection in the population and alternately dominate the population (see Fig. 2c, d). At this stage, the game payoff of all the strategies starts to increase beyond zero. I call this periodic orbit the black queen orbit.

The picture depicted above is the typical behavior of the model for large enough values of the cost. To see this, in Fig. 3a, I plot the phase diagram of the model in the $c_g - r$ plane. Here, the frequency of cooperators in the population, $\rho_C = \rho_C^1 + \rho_C^2$, is color plotted as well (see Supplementary Fig. 1 for the frequency of different strategies). Red dashed lines show the boundary of the region where the system settles in a periodic orbit. For high costs, as $r$ increases, the system shows a series of successive cross-overs from a defective fixed point to the red queen periodic orbit, black queen periodic orbit, and finally a cooperative fixed point. On the other hand, for small costs, the system possesses a bistable region where both the red queen and black queen periodic orbits (or a partially cooperative fixed point and black queen periodic orbit to the left of the red dashed line in the bistable region) are stable, and the system shows a discontinuous transition between these two orbits. Orange circles show the lower boundary of the bistable region, below which the black queen orbit is unstable. Its upper boundary, above which the red queen orbit becomes unstable, is plotted by red squares, in Fig. 3. The transition between the two periodic orbits becomes a continuous transition at a single critical point (see Supplementary Fig. 4).

Examination of the overall cooperation in the population shows that an entrance cost has a contrasting effect on population cooperation for large and small enhancement factors. An entrance cost keeps free-riders away from a costly institution. This fact makes the relative frequency of cooperators to defectors higher in the costly institution than that in the free institution. To see that defectors are less likely to join the costly institution, I plot the difference between the probabilities that an individual in the costly institution is a cooperator and the probability that an individual in the free institution is a cooperator, $\gamma = \rho_C^1/(\rho_C^1 + \rho_D^1) - \rho_C^2/(\rho_C^2 + \rho_D^2)$ in Fig. 3b, where it can be seen it is always positive. Intuitively, as a costly defector's payoff in a group with $n_C^1$ cooperators and $n_D^1$ other defectors is equal to $r n_C^1/(n_C^1 + n_D^1) - c_g$, a costly defector can reach a positive payoff only when $n_C^1$ is large. Otherwise, costly defectors are better off hedging the risk of obtaining a negative payoff by joining the free institution, where their payoff is necessarily non-negative. Consequently, the expected number of cooperators in the costly institution, $\rho_C^1 g$, sets a bound for the frequency of costly defectors. This fact increases a costly institution's profitability, especially for small enhancement factors, and positively impacts cooperation in the population. On the other hand, for high enhancement factors, a large entrance cost is detrimental to cooperation. This is because, although the frequency of defectors in the costly institution remains close to zero, fewer individuals are willing to choose a costly institution with a high cost. This increases the effective size of the free institution and the mixing between cooperators and defectors in the free institution. Since defectors can better exploit cooperators in well-mixed groups, the increased

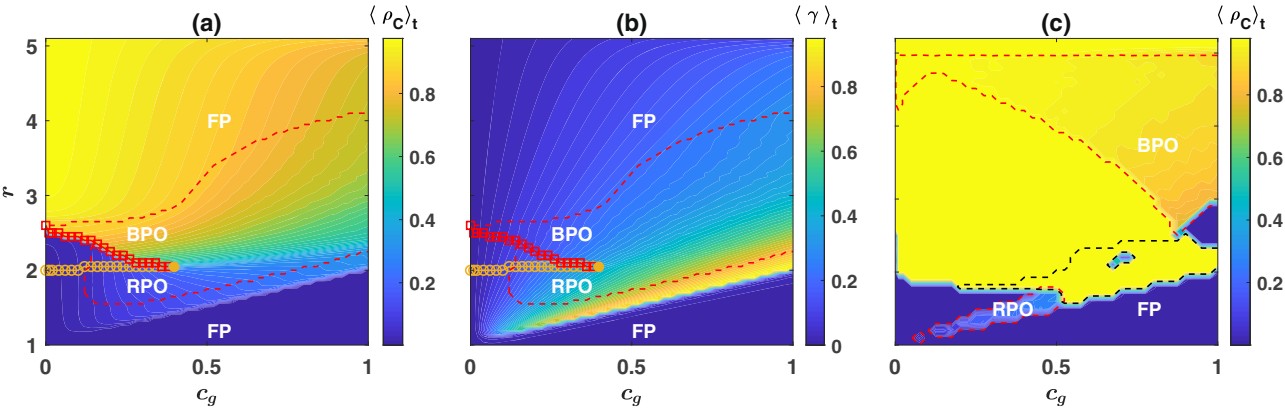

**Fig. 3 Evolution of cooperation. a** Time average total frequency of cooperators, $\rho_C = \rho_C^1 + \rho_C^2$ in the $r - c_g$ plane is color plotted. The dynamics can settle in fixed point (FP) (small and large enhancement factors), or two different periodic orbits, red queen periodic orbit (RPO) where cooperation only in the costly institution evolve and black queen periodic orbit (BPO) where cooperation in both institutions evolve. **b** Time average difference between the probability that an individual in the costly institution is a cooperator from the probability that an individual in the free institution is a cooperator, $\gamma = \rho_C^1/(\rho_C^1 + \rho_D^1) - \rho_C^2/(\rho_C^2 + \rho_D^2)$. Individuals are more likely to be cooperators in a costly institution. **c** The time average total frequency of cooperators in the $r - c_g$ plane under pure selection dynamic ($\nu = 0$). Red queen and black queen periodic orbit can occur for, respectively, small and large enhancement factors. In other regions, the dynamics settle in a fixed point where either non-costly defectors (small enhancement factors), costly cooperators (inside the region marked with dashed black line), or non-costly cooperators survive. Parameter values: $g = 5$, and $\pi_0 = 2$. In (**a**, **b**) $\nu = 10^{-3}$, and in (**c**) $\nu = 0$. In (**a**, **b**) the replicator dynamic is solved for 8000 time steps, and the time average is taken over the last 2000 steps. In (**c**) the replicator dynamic is solved for 200,000 time steps, and the time average is taken over the last 150,000 steps.

mixing between cooperators and defectors in the free institution hinders cooperation.

As shown in the Supplementary Note 3, while the phenomenology of the model remains the same for lower mutation rates, lower mutation rates increase the size of the region where the dynamics settle in a periodic orbit (see Supplementary Fig. 5). Regarding the dependence of the dynamics on the mutation rate, an interesting case is the zero mutation rate, where selection is the sole driver of the dynamics. The time average cooperation for zero mutation rate, starting from an initial condition where all the strategies are equal, is plotted in Fig. 3c (See Supplementary Fig. 6 for the frequency of different strategies). Both the red queen (for small enhancement factors) and the black queen (for large enhancement factors) periodic orbits are observed in this case. However, for zero mutation rate, both the amplitude and period of fluctuations increase: The fluctuating dynamics go through periods where one of the surviving strategies reaches a frequency close to 1 only to be later replaced by another strategy (see Supplementary Fig. 7). The dynamics can also settle in different fixed points. For $c_g = 0$, depending on the enhancement factor, either cooperators or defectors in both institutions survive in equal densities. For nonzero $c_g$, however, only one of the strategies survives. For small enhancement factors, non-costly defectors dominate the population. For larger enhancement factors, either costly cooperators (the region marked with a dashed black line) or non-costly cooperators dominate the population.

In the Supplementary Note 2, I consider a case where the two institutions have different productivities, i.e., different enhancement factors, and show that similar phases are at work in this case (see the Supplementary Figs. 2 and 3). For instance, I show that a large entrance cost destabilizes full defection, removes the system's bistability, and ensures the evolution of cooperation starting from all the initial compositions of the population. In addition, I study the continuous replicator dynamics and show similar phenomenology is at work in this case (see Supplementary Notes 1.4 and 4, and Supplementary Figs. 8 and 9).

Finally, I note that a similar phenomenology is at work in a context where instead of a costly and a cost-free institution, two costly institutions interact. To see this, assume institution 1 has a cost $c_g$ and institution 2 has a cost $c_g^0$. Without loss of generality, assume $c_g > c_g^0$. Writing $c_g = (c_g - c_g^0) + c_g^0$, it is easy to see that it is possible to absorb $c_g^0$ in the base payoff $b$ (as all the individual pay a cost $c_g^0$ irrespective of their institution choice). Thus, the model is equivalent to a context where resource 2 has zero cost, resource 1 has a cost of $c_g - c_g^0$, and all the individuals receive a shifted base payoff of $b - c_g^0$ (see Supplementary Note 5 and Supplementary Fig. 10).

**Structured population.** In contrast to the well-mixed population, the model shows no bistability in a structured population, and the fate of the dynamics is independent of the initial condition. To see why this is the case, I note that in a well-mixed population, a situation where all the individuals are defectors, and randomly prefer one of the two institutions, is the worst case for the evolution of cooperation, as in this case, mutant cooperators are in a disadvantage in both institutions. However, in a structured population, starting from such an initial condition, blocks of defectors, most of whom prefer the same institution, form due to spatial fluctuations. A mutant cooperator who prefers the minority institution in these blocks obtains a high payoff and proliferates. This removes the bistability of the dynamics in a structured population.

To study the model's behavior in a structured population, I perform simulations starting from an initial condition in which all the individuals are defectors and prefer one of the two institutions at random. The model shows similar behavior in a structured population to that in a well-mixed population. This can be seen in Fig. 4a–d, where the densities of different strategies are color plotted in the $c_g - r$ plane (see Supplementary Note 6 and Supplementary Figs. 11 and 12 for further details). As was the case in a well-mixed population, cooperation does not evolve for too small values of $r$. As $r$ increases beyond a threshold, cooperation does evolve in the costly institution but not in the free institution. In this region, for a fixed enhancement factor, an optimal cost, approximately equal to $c_g = r - 1$, optimizes the cooperation in the population. On the other hand, cooperation in

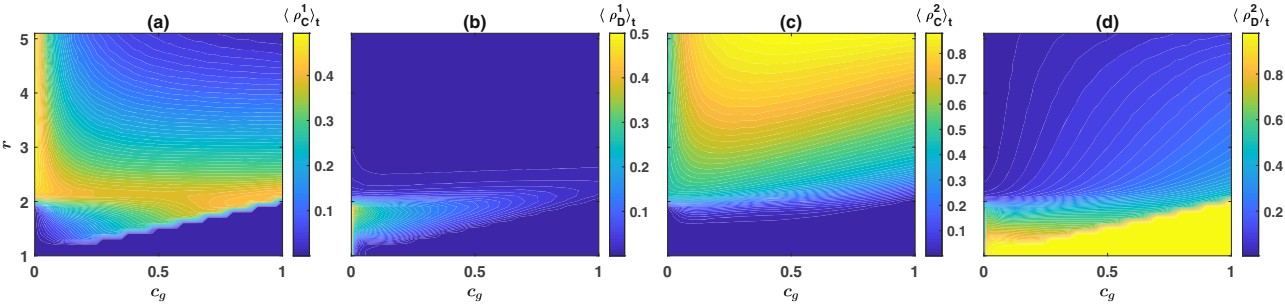

**Fig. 4 The frequency of different strategies in the $c_g - r$ plane in a structured population.** The time average frequencies of costly cooperators (**a**), costly defectors (**b**,) non-costly cooperators (**c**), and non-costly defectors (**d**) in the $c_g - r$ plane are color plotted. The system shows a red queen dynamic in which cooperators only in the costly institution survive in large numbers (for smaller enhancement factors), or a black queen dynamic, where cooperators in both institutions survive and help each other to suppress defection (for larger enhancement factors). Parameter values: $g = 5$, $nu = 10^{-3}$, and $\pi_0 = 2$. The population resides on a $200 \times 200$ first nearest neighbor square lattice with von Neumann connectivity and periodic boundaries. The simulation is performed for 5000 time steps starting from an initial condition in which all the individuals are defectors and prefer one of the two institutions at random. The time average is taken over the last 2000 steps.

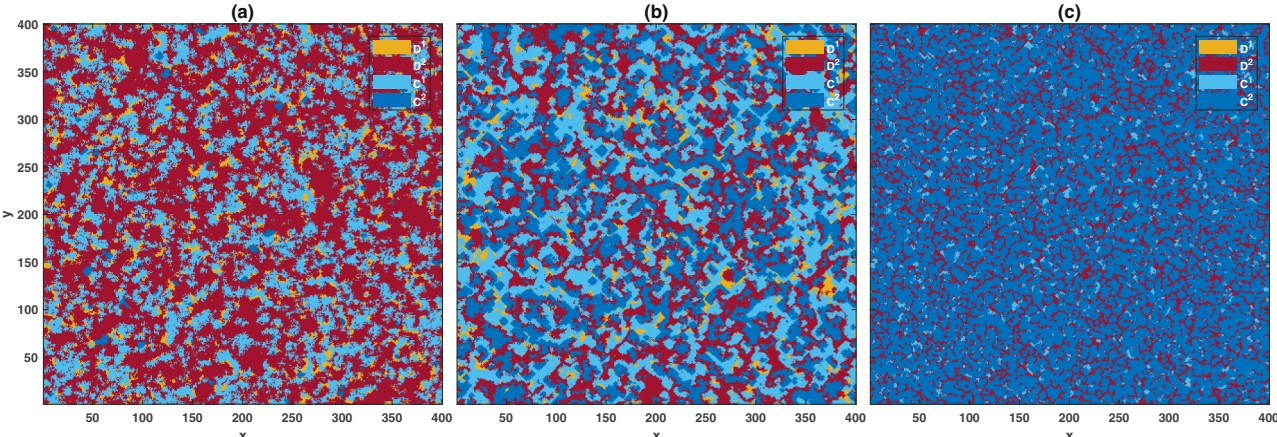

**Fig. 5 Snapshots of the population in the stationary state for different parameter values.** Different strategies are color codded (legend). In (**a**,) $r_1 = 1.7$, $r_2 = 1.7$, in (**b**,) $r_1 = 3.5$ and $r_2 = 3.5$, and in **c**, $r_1 = 3$, and $r_2 = 1.8$. In all the cases $c_g = 0.6$. For small enhancement factors (**a**), the red queen dynamics in which cooperators only in the costly institute survive in large numbers occur. For larger enhancement factors (**b**), the black queen dynamics in which cooperators in both institutions survive and help each other suppress defection occur. By increasing the enhancement factors (**c**), non-costly cooperators dominate. However, a small frequency of costly cooperators survives and purge the population from defectors by moving along the bands of non-costly defectors. Here, individuals reproduce with a probability proportional to the exponential of their payoff with a selection parameter equal to $\beta = 5$. The population resides on a $400 \times 400$ square lattice with von Neumann connectivity and periodic boundaries. Parameter values: $g = 5$ and $\nu = 10^{-3}$.

both the costly and the free institutions evolves for large enhancement factors. In this region, increasing the cost can slightly increases defection in the free institution and have a detrimental effect on the evolution of cooperation, but not as much as it does in a well-mixed population.

Instead of periodic orbits observed in the well-mixed population, on a spatial structure the model's dynamic is governed by the cyclic dominance of different strategies through spatiotemporal fluctuations manifested by traveling waves. In Fig. 5, I present snapshots of the population's stationary state in different phases. In this figure, I consider a model in which individuals reproduce with a probability proportional to the exponential of their payoff, $\pi$, times a selection parameter, $\beta$, $\exp(\beta\pi)$ (see the Supplementary Note 1.3), with $\beta = 5$. The situation in the model where individuals reproduce with a probability proportional to their payoff is similar. In Fig. 5a, I have set $r_1 = r_2 = 1.7$, and $c_g = 0.6$. This phase corresponds to the red queen periodic orbit in the well-mixed population case. Here, the majority of the population are non-costly defectors. Costly cooperators experience an advantage over the former and can proliferate in the sea of non-costly defectors. However, costly cooperators are

vulnerable to both costly defectors and non-costly cooperators. The former can only survive in small bands around costly cooperators, as they rapidly get replaced by non-costly defectors once they eliminate costly cooperators. This phenomenon shows that spatial competition between defectors with differing institution preferences can positively impact the evolution of cooperation. Non-costly cooperators, in turn, can survive by forming compact domains where they reap the benefit of cooperation among themselves. However, as the effect of network reciprocity is too small to promote cooperation in this region, non-costly cooperators get eliminated by non-costly defectors once costly cooperators are out of the picture. Consequently, the system's dynamic is governed by traveling waves of costly cooperators followed by small trails of costly defectors and non-costly cooperators in a sea of non-costly defectors (see the Supplementary Video, SV.1[56], and Supplementary Note 7 for an illustration of the dynamics in this regime).

Figure 5b shows a snapshot of the population for $r_1 = r_2 = 2.2$. This phase corresponds to the black queen periodic orbit in a well-mixed population. In this phase, cooperators in both the costly and free institutions evolve. Cooperators are vulnerable to

defectors in their institution and lose their territory to defectors of similar type. Defectors are in turn vulnerable to cooperators in their opposite institution and are replaced by them. Consequently, the dynamic of the model is governed by traveling waves of cooperators, chased by defectors of similar type, who are in turn extincted by cooperators of the opposite type. Thus, while cooperators of different types on their own either can not survive (in the case of non-costly cooperators) or are doomed to a winnerless competition with defectors (in the case of costly cooperators), they complement each other to efficiently suppress defection in the population (see the Supplementary Video, SV.2[56], for an illustration of the dynamics in this regime).

Another manifestation of functional complementation between cooperators of different types can be seen in the regime of large enhancement factors. An example of this situation is plotted in Fig. 5c. Here, $r_1 = r_2 = 3.5$ and $c_g = 0.6$. In this region, non-costly cooperators dominate the population. However, non-costly defectors can survive in small bands in the sea of non-costly cooperators. While at a disadvantage in the sea of non-costly cooperators, costly cooperators beat non-costly defectors. Consequently, small blocks of costly cooperators are formed within the bands of non-costly defectors. These blocks of costly cooperators move along the bands of non-costly defectors and purge the population from non-costly defectors. In this way, although costly cooperators exist only in small frequency, they play a constructive role in helping non-costly cooperators to dominate the population.

In summary, the analysis of the spatial patterns reveals that competition or synergistic relation between individuals with different institution preferences plays an essential role in the evolution of cooperation in the system. Defectors with different institution preferences always appear as competitors who compete over space. By eliminating each other, they play a surprisingly constructive role in the evolution of cooperation. Cooperators, on the other hand, while having direct competition over scarce sites, can also act synergistically and help the evolution of cooperation in their opposite institution since they can eliminate defectors in their opposite institution. In this way, by purging defectors with an opposite game preference, cooperators help fellow cooperators with an opposite game preference. Consequently, cooperators with different game preferences can engage in a mutualistic relation to efficiently suppress defection in the population.

Finally, as shown in the Supplementary Note 6, the spatial model shows similar phases in the case where the two public resources have heterogeneous profitability, that is, when $r_1 \neq r_2$ (see the Supplementary Fig. 12).

## Discussion

The question of how participation cost affects the evolution of cooperation had been considered in the context of binary interactions (prisoner's dilemma), for instance, in the context of indirect reciprocity[57], structured populations[58–60], and in repeated group interactions[61,62]. Some previous work shows that when interactions are obligatory, participation cost can have a detrimental effect on the evolution of cooperation[57,58]. As I have shown, the advantage of a costly institution becomes apparent when an alternative free institution accompanies it. In such contexts, while defectors predominantly join a free institution, cooperators can reach a higher payoff by working cooperatively in a costly institution, despite paying a participation cost with no direct benefit. Furthermore, the existence of a costly institution can have a positive impact on the evolution of cooperation in the free institution. In a well-mixed population, this is brought about by the fact that non-costly defectors can be eliminated by costly

cooperators. While this mechanism can be at work to promote cooperation in a well-mixed population, it is further strengthened in the presence of population structure. In a structured population, costly cooperators, while at a disadvantage in the presence of non-costly cooperators or costly defectors, can beat non-costly defectors. Consequently, they can grow in domains of non-costly defectors, and by purging the population from non-costly defectors, they help the evolution of non-costly cooperators.

While in many cases, individuals engage in a heterogeneous collective action dilemma, in which they need to choose between different collective action tasks, this context had been eluded theoretical examination until very recently. By considering such heterogeneous public goods dilemmas, ref. [45] shows having a choice between different institutions with differing productivity can promote cooperation. In this context, an institution with low productivity is less likely to attract defectors and thus behaves as a shelter, in which cooperators can shelter to avoid defectors. Here, I have studied the effect of an entrance cost in heterogeneous public goods. While a costly institution has similarities with a low productivity institution, in that it behaves as a cooperation niche, it is more effective in solving collective action problems, especially for small enhancement factors where, in the absence of a participation cost, cooperation does not evolve. In addition, a participation cost can remove the bistability of the dynamics, an ineliminable feature of cost-free heterogeneous public goods, and ensures the evolution of cooperation starting from all the initial conditions. This observation is of importance, as it implies that a single mutant cooperator can take over a population of defectors in the presence of sufficiently high participation costs. Furthermore, as we argued, a similar phenomenology is at work in a situation when two costly institutions with differing costs coexist.

Analysis of the model in a structured population reveals interesting insights about how having a choice between different resources can help the evolution of cooperation. In a structured population, defectors with different game preferences are in direct competition over scarce spatial resources. By undermining each other, they help the evolution of cooperation in their opposite institution. Cooperators with opposite game preferences are in direct competition as well. However, as cooperators can beat defectors with the opposite game preference and grow in their domain, they help their rival cooperators with a different game preference to flourish. Consequently, a mutualistic relation between cooperators with opposite game preferences emerges, which helps the evolution of cooperation. This is reminiscent of a black queen dynamic—suggested as a pathway towards the evolution of dependencies between organisms in the presence of antagonistic relations[36,55]—in which different types depend on each other for performing the vital function of suppressing the cheaters. Despite being in direct competition, a stable polymorphism between cooperators with different game preference emerges. One can think of this polymorphism as an intermediate step towards the evolution of division of labor, in which cooperators with different game preferences crucially depend on each other to get rid of cheaters in their community[36,63]. While this black queen dynamic is observed for high profitability of the public goods, in the other extreme of low profitability, where the conditions are more hostile for the evolution of cooperation, a different dynamic, the red queen dynamic, is observed. In this regime, cooperation is maintained only in a costly institution. Rather than mutualistic relation between cooperators with different game preferences, cooperation in this regime crucially depends on the existence of an entrance cost that keeps free-riders away from a costly institution.

Empirical examples of a situation where individuals have a choice between different public resources to perform a collective action task appear to abound in human societies. Some examples

include competition between political parties, firms, and associations, and team incentives within firms[40]. A choice between geographic locations such as cities[42,43], or having a choice between different collective action tasks such as communal hunting, gathering, or communal agriculture[44] provide other examples. As our analysis suggests, the existence of a costly alternative, or more generally, heterogeneous entrance costs, can positively impact the evolution of cooperation both in the costly resource and its competing resources. In the case of large-scale human institutions, such a participation cost can be costly signaling of a common goal, for example, in the form of a donation to charity[64], or transaction cost associated with the search, bargaining, or monitoring to maintaining the commons[65]. An interesting example is food acquisition in hunter-gatherer societies. Hunting or gathering activities are subject to different costs, such as the acquisition of the resources required for hunting and the heterogeneous risk involved in hunting different prey or gathering nutritional resources[66]. Indeed, given the high cost and low reward of some hunting efforts, the question arises what function does such costly activities, often regarded as costly signals, perform in many small-scale human societies[67,68]? While the relation between such costly signals and cooperation has been noted[69,70], our theory can provide a novel perspective on the evolutionary advantage of such costly cooperative activities. According to our theory, a group seeking a high-risk, costly collective activity can perform better due to higher cooperation induced by the high cost, which, in turn, may have a positive impact on alternative low-cost collective actions. Another related example where the mechanism suggested here can be at work in human groups is costly signals of social status, which can be used for choosing partners and admitting others in the group[66,68]. By showing that a cost can unexpectedly enhance cooperation and productivity, our theory suggests a novel explanation for the evolution of such costly signals. According to this viewpoint, one can think of the cost paid to join the costly institution as a costly signal with no immediate benefit, such as public display of generosity, donation, or risk-taking behavior, required to be admitted in the group.

Examples of a situation where individuals have a choice between different resources appear to exist in other biological populations as well. An example is the fission-fusion dynamics observed in different animal groups[3,5,71], or even hunter-gatherer human societies[41]. In many such cases, sub-groups are formed within a group to perform a collective action task, such as cooperative hunting[3,4], or cooperative defending[3]. Another example is resource competition in microorganisms[1,2]. An example is siderophore production in microbial populations[36–39]. Bacteria use siderophores to uptake iron. Microbial strains produce a diverse set of siderophores. Microbes can use the siderophore produced by a microbe with matching receptors. Consequently, bacteria face both exploitations by the cheats (with the same siderophore production system) and competition with other strains with different iron-uptake systems, i.e., with a different public goods preference[39]. This system seems an interesting example where the phenomenology suggested by our model can be at work. For instance, arguablely, maintaining different iron-uptake systems have heterogeneous costs. As our analysis suggests, the existence of such costs can be an important mechanism for the promotion of cooperation in these populations. Furthermore, our analysis suggests different regimes, a winner-less red queen dynamic or a black queen polymorphic coexistence between different strains, can be at work under different environmental conditions. While siderophore production has been subject to growing interest in recent years, and the evolutionary arms race between different microbial strains is noted as a salient feature of these systems[36–39], whether such

different regimes can be at work, and to what extent cost heterogeneity is appealed to solve collective action problems in these populations, as well as in other biological populations where multiple public goods exist and interact, remain to be addressed by future empirical research.

## Methods

**The replicator dynamics.** The model can be described in terms of the replicator-mutators equation[72]. This equation reads as follows:

$$\rho_x^i(t+1) = \sum_{y,j} \nu_{y,j}^{x,i} \rho_y^j(t) \frac{\pi_y^j(t)}{\sum_{z,l} \rho_z^l(t)\pi_z^l(t)}. \qquad (1)$$

Here, $x$, $y$, and $z$ refer to strategies and can be either $C$ or $D$, and $i$, $j$, and $l$ refer to the public resources which can be 1 or 2. $\nu_{y,j}^{x,i}$ is the mutation rate from a strategy profile that prefers public resource $j$ and plays strategy $y$ to a strategy combination that prefers public resource $i$ and plays strategy $x$. These can be written in terms of mutation rates as follows:

$$\begin{cases} \nu_{y,j}^{x,i} = 1 - 2\nu + \nu^2, & \text{if } (i = j \text{ and } x = y) \\ \nu_{y,j}^{x,i} = \nu - \nu^2, & \text{if } (i = j \text{ and } x \neq y) \text{ or } (i \neq j \text{ and } x = y) \\ \nu_{y,j}^{x,i} = \nu^2 & \text{if } (i \neq j \text{ and } x \neq y) \end{cases} \qquad (2)$$

In eq. (1), $\pi_y^j$ is the expected payoff of an individual who prefers public resource $j$ and plays strategy $y$. These terms can be written by averaging a focal individual's payoff with game preference $j$ (for $j = 1$ and 2) in a group composed of $n_C^j$ cooperators and $n_D^j$ defectors who prefer public resource $j$, over all possible group configurations. In this way, I have the following equations for the payoffs:

$$\pi_C^1 = \sum_{n_D^1=0}^{g-1-n_C^1} \sum_{n_C^1=0}^{g-1} cr_1 \frac{1+n_C^1}{1+n_C^1+n_D^1}$$
$$(1-\rho_C^1-\rho_D^1)^{g-1-n_C^1-n_D^1} \rho_D^{1\,n_D^1} \rho_C^{1\,n_C^1} \binom{g-1}{n_C^1,n_D^1,g-1-n_C^1-n_D^1} - c - c_g + \pi_0,$$

$$\pi_D^1 = \sum_{n_D^1=0}^{g-1-n_C^1} \sum_{n_C^1=0}^{g-1} cr_1 \frac{n_C^1}{1+n_C^1+n_D^1}$$
$$(1-\rho_C^1-\rho_D^1)^{g-1-n_C^1-n_D^1} \rho_D^{1\,n_D^1} \rho_C^{1\,n_C^1} \binom{g-1}{n_C^1,n_D^1,g-1-n_C^1-n_D^1} - c_g + \pi_0.$$

$$\pi_C^2 = \sum_{n_D^2=0}^{g-1-n_C^2} \sum_{n_C^2=0}^{g-1} cr_2 \frac{1+n_C^2}{1+n_C^2+n_D^2}$$
$$(1-\rho_C^2-\rho_D^2)^{g-1-n_C^2-n_D^2} \rho_D^{2\,n_D^2} \rho_C^{2\,n_C^2} \binom{g-1}{n_C^2,n_D^2,g-1-n_C^2-n_D^2} - c + \pi_0, \qquad (3)$$

$$\pi_D^2 = \sum_{n_D^2=0}^{g-1-n_C^2} \sum_{n_C^2=0}^{g-1} cr_2 \frac{n_C^2}{1+n_C^2+n_D^2}$$
$$(1-\rho_C^2-\rho_D^2)^{g-1-n_C^2-n_D^2} \rho_D^{2\,n_D^2} \rho_C^{2\,n_C^2} \binom{g-1}{n_C^2,n_D^2,g-1-n_C^2-n_D^2} + \pi_0.$$

In this equation, $cr_1 \frac{1+n_C^1}{1+n_C^1+n_D^1} - c$ in the first equation is the payoff of a cooperator who prefers the public resource 1, and $cr_1 \frac{n_C^1}{1+n_C^1+n_D^1}$ in the second equation is the payoff of a defector who prefers public resource 1. $(1-\rho_C^1-\rho_D^1)^{g-1-n_C^1-n_D^1} \rho_D^{1\,n_D^1} \rho_C^{1\,n_C^1} \binom{g-1}{n_C^1,n_D^1,g-1-n_C^1-n_D^1}$, is the probability that such a group composition occurs. Here, $\binom{g-1}{n_C^1,n_D^1,g-1-n_C^1-n_D^1} = \frac{(g-1)!}{n_C^1! n_D^1! (g-1-n_C^1-n_D^1)!}$ is the multinomial coefficient and is the number of ways that $n_C^1$ cooperators and $n_D^1$ defectors who prefer game 1 can be chosen among $g - 1$ group-mates of a focal individual. Summation over all the possible configurations gives the expected payoff of a cooperator, or defector who prefers resource 1. As resource one is a costly institution, the individual needs to pay a participation cost, $c_g$. Finally, a base payoff of $b$ is added to all the payoffs. It is easy to derive the expected payoff of those who prefer public resource 2, using a similar argument.

In the Supplementary Notes 1.3, and 8, a model where individuals reproduce with a probability proportional to the exponential of $\beta\pi$, where $\pi$ is the payoff, and $\beta$ is a selection parameter is considered. The replicator dynamics of such a model are derived using similar steps, and the model shows similar behavior to the model introduced in the main text (See Supplementary Figs. 13–16).

**Statistics and reproducibility.** Figures 1–3 result from numerical solution of the replicator dynamics. The initial condition is a uniform initial condition in which all the strategies' initial densities are equal ($\rho_C^1 = \rho_D^1 = \rho_D^2 = \rho_D^2 = 0.25$). In Fig. 1 the replicator dynamics is solved for $T = 9000$ time steps, and an average over the last 2000 time steps is taken. In Fig. 3a, b, the replicator dynamics is solved for $T = 8000$ time steps, and an average over the last 2000 time steps is taken, and in Fig. 3c it is solved for $T = 200,000$ time steps and time averages are taken over the last 150,000 steps. The simulations in a structured population presented in Fig. 4

are performed for 5000 time steps starting from an initial condition in which all the individuals are defectors and prefer one of the two institutions chosen randomly. The time averages are taken over the last 2000 time steps. As the model shows no bistability in a structured population, different initial conditions result in the same picture.

To derive the phase diagrams, I solve the replicator dynamics starting from different initial conditions. In the mono-stable region, the stationary state of the dynamics is the same for different initial conditions. This allows us to derive the boundary of the cyclic phase (the white line) using any of the initial conditions. To determine the boundary of bistability, I note the most facilitative initial condition for the evolution of cooperation is the one in which all the individuals are cooperators and prefer the costly institution. The lower boundary of the bistable region above which the cooperators can survive is determined starting from this initial condition. The worst initial condition for the evolution of cooperation is the one in which all the individuals are defectors and prefer the two institutions in similar frequencies. The replicator dynamics solutions starting from this initial condition give the upper boundary of the bistable region above which the defective state becomes unstable.

The agent-based simulations are performed according to the model definition. In the well-mixed population, at each generation, the population is decomposed into $N/g$ groups formed at random. Each agent has two binary traits that determine game preference (1 or 2) and strategy ($C$ cooperation or $D$ defection). Agents gather payoffs by playing a public goods game in the institution of their choice. In addition, those who choose the costly institution pay the entrance cost of $c_g$. After gathering payoffs, selection occurs. In the selection stage, for each agent in the next generation, one of the agents in the past generation is chosen as a parent with a probability proportional to its payoff. The offspring inherits the strategy of its parent subject to mutations. Mutations in the game preference and strategy occur independently and each with a probability $v$. In the case of a mutation, the corresponding variable is flipped to its opposite value.

In the structured population, agents reside on a nearest neighbor lattice with Von Neumann connectivity and periodic boundaries. Each agent is a member of 5 groups, each centered around itself or one of its neighbors. Agents in each group enter the public goods game of their choice and gather payoffs (as explained in the well-mixed population case). For the selection stage, I consider a death-birth process in which each agent adopts the strategy of an agent in its extended neighborhood with a probability proportional to its payoff. The extended neighborhood of an agent is defined as the focal agent together with its neighbors on the network. As in the well-mixed population case, I consider a synchronous dynamic or non-overlapping generation model where the whole population is updated at the same time.

**Reporting summary**. Further information on research design is available in the Nature Research Reporting Summary linked to this article.

## Data availability
The datasets generated during and/or analyzed during the current study are available in the ZENODO repository, https://doi.org/10.5281/zenodo.5541182 [73].

## Code availability
All the computer codes used in this study were written in MATLAB 2019a or 2017b. The computer codes used to prepare figures were written in MATLAB 2017b. All the MATLAB codes are publically available at the ZENODO repository, https://doi.org/10.5281/zenodo.5541182 [73]

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

## Acknowledgements
The author acknowledges funding from Alexander von Humboldt Foundation in the framework of the Sofja Kovalevskaja Award endowed by the German Federal Ministry of Education and Research. Three anonymous reviewers are acknowledged for fruitful suggestions.

## Author contributions
M.S. designed the research. M.S. performed the research. M.S. wrote the paper.

## Funding

## Competing interests
The author declares no competing interests.
