## [Transparent Peer Review File · Communications Biology]

Reviewers' comments:

Reviewer #1 (Remarks to the Author):

The paper studies a model where individuals follow cooperative or defective strategies in the classical public goods game. However, differing to the traditional models, individuals follow another strategy in parallel: either they join to the group with costly institution or the cost free group. That is, institution cost suffers both the cooperators and defectors. The author pointed out that this kind of heterogeneity in the game facilitates the cooperation, and can lead to the coexistence of cooperators and defectors in the well-mixed and the spatially structured model.

The problem of maintaining of cooperation in public goods situation is one of the classical problem of game theory and evolutionary biology. There are numerous solutions to the problem. This present paper suggests an alternative, less studied way to maintain cooperation.

While the problem is important and the studied mechanism is interesting, the implementation of the project is problematic for numerous points. Below I collected my major and the minor points.

Major points

1. The paper is not clear enough in several points. The introduction should give examples (see my point later) where the suggested mechanism can have crucial effect on maintaining group cooperation. The first paragraph is too compact and consequently it has huge logical jumps. There are several papers which focuses on the role of individual or environmental heterogeneity on the evolution of cooperation (see e.g. Kun Á., Dieckmann, U. 2013 Resource heterogeneity can facilitate cooperation Nature Communications 4: 2453, Stilwell, P., O'Brien, S., Hesse, E., Lowe, C., Gardner, A. and Buckling, A. (2020), Resource heterogeneity and the evolution of public goods cooperation. Evolution Letters, 4: 155-163). I suggest to study these papers and references therein and cite them accordingly.

2. The model description is not detailed enough. For example, it is not determined clearly in the main text how the alternative groups are formed. Similarly, it should be described in more detail how the agent based models work, both in the well-mixed and the spatial model.

3. The studied model is only a variant of the model published by the author recently (Salahshour PLOS Comp. Biol 2021). He investigates the situation when there is a constant cost for joining to one of the cooperating group in the present ms, while he studied the case when the maximal reward of the cooperation is different in different environments in the PLOS Comp. Biol. paper. The two models are very similar in other aspects. The author should analyse the similarities and differences between the two models both in the introduction and in the discussion.

4. The paper presents the dynamical behaviour of the system in detail (see Fig. 3-5) which was already investigated in the PLOS paper in a similar model. I think the main result is the coexistence of cooperators and defectors as a consequence of individuals can join to different groups. These dynamical issues are less important.

The author should explain convincingly what is the reason of higher level of cooperation and coexistence of cooperators and defectors is typical in this system. I note here that if one considers a single environment with the same simple public goods game, then either cooperators ($r/g > c$) or the defectors ($r/g < c$) win the competition. The situation is characteristically different in the present model, but why? A deeper mathematical argumentation would be convincing.

5. My other general problem with the paper is, that I don't think this kind of mechanism is common in biology and similarly I think it is rare in human population. The author mentions some examples where individuals can join alternative cooperative groups at the end of the discussion, but there is no cost of joining to the group or having an extra cost of being the member of one of the group in these examples. Further, we know that other mechanisms maintain the cooperation in the presented examples (private benefit of the cooperators, reputation, punishment, ostracism, partner preference, etc.).

Minor points

1. The abstract should emphasize that this is a theoretical work. To say that individual have a choice between the groups is a bit misleading. Choosing a group is a heritable strategy.
2. There are numerous typos in the text.
3. Figure 1: Subfigures should be designated. c_g emerges here first without defining it before. You mentioned replicator dynamics. I suggest to refer to the method section here.
4. It is not clear what is the role of including mutation among the strategies. It would be interesting to study the dynamics when only selection is allowed (naturally all ρ -s should be bigger than zero initially).
5. It is not clear what does happen if actually only one cooperator individual is present in a group? I note here, that the story would be totally different if the reward is a nonlinear function of the number of cooperators.
6. I would change "mixed population" expression to well-mixed.
7. The experienced periodic behaviour can be simply the consequence of using discrete time dynamics in replicator dynamics. Using the differential equation version can easily eliminate the dynamical instability.
8. Results, first sentence: There is no analytical solution of the problem. The author solved the analytical model by numerical integration. For that reason the sentence "The result of the simulations in finite populations..." in the same paragraph is not precise. The numerical simulations of the analytical model and the numerical simulations of the agent based model lead to similar result This will be more adequate.
9. Results, second paragraph: The explanation of the result is absolutely not convincing. We need some mechanistic or at least intuitive explanation.
10. Page 7. 4b should be 4d.
11. Page 9. Fig. 8. It is not clear why a different update rule (exponential payoff) is used here.
12. Page 9. last par. It is rather trivial that costly cooperator dominates if $r_1 - c_g > r_2$.

Reviewer #2 (Remarks to the Author):

In "Evolution of cooperation in costly institutes" authors show that in a situation where individuals have a choice between a costly institute and a free institute to perform a collective action task, the existence of a participation cost promotes cooperation in the costly institute. This is counterintuitive, because despite paying for a participation cost, costly cooperators can outperform defectors, who predominantly join a free institute. What is more, cooperation is thereby promoted also in the free institute. Research also shows that in a structured population a mutualistic relation between cooperators with different institute preferences emerges, which also helps the evolution of cooperation.

How and why cooperation emergence in public goods dilemmas is an intensely investigated subject with obvious practical ramifications. Methods of computational biology, statistical physics, and network science have been applied successfully and with much effect recently to shed light on the problem from many different perspectives, and also to outline many different ways on how solutions in terms of enhanced cooperation could be obtained. In this sense, the study addresses a relevant setup, and it also delivers results that might be of interest to the readership of Communications Biology.

I have quite enjoyed reading this manuscript, but I also have a couple of comments that require revision.

1) In the first place, I am missing an intuitive explanation of these very counterintuitive results. It is already unexpected that the existence of a participation cost promotes cooperation in the costly institute. But then also cooperation is promoted in the free institute. Such rather bold conclusions require the identification of the underlying mechanism, and I would ask the author to spell this out to me in response to this comment, and then also in the manuscript. Preferably already in the abstract as

much as this is possible, but then also definitively during the results and in the discussion.

2) In terms of closely related research related to cooperation in the public goods game and institutions, there are two research papers that have been overlooked, namely Phase diagrams for the spatial public goods game with pool punishment, *Phys. Rev. E* 83, 036101 (2011), and Sustainable institutionalized punishment requires elimination of second-order free-riders, *Sci. Rep.* 2, 344 (2012). In both cases institutions are referred to as pool strategies, but the concept behind the considerations is the same. Essentially a governing body that is above an individual strategy that exerts some form of punishment, reward, or similar.

3) It would also improve the paper if the figure captions would be made more self contained. In addition to what is shown for which parameter values, one could also consider a sentence or two saying what is the main message of each figure.

Reviewer #3 (Remarks to the Author):

This paper analyzes a model where individuals have a choice of which institute (aka game) to join, one of which has a participant cost and one does not, and whether to cooperate or not (binary decision). The two traits mutate and evolve independently. Individuals inherit or copy the traits according to the relative payoff success of individuals in preceding generation, and the population can be well mixed or spatially structured on a network/lattice. An analysis with institutions that vary in quality is also included. The overall conclusions are that a costly institute can favour cooperation, under certain parameters (when profitability, or enhancement factor, not too high).

Overall, this paper is of quite broad interest, as it deals with the evolution of cooperation, and is particularly orientated towards human type problems and solutions to cooperation. I also found the model description well written. However, I unfortunately find the whole paper quite difficult to digest, primarily the results, and some key points do not seem to be sufficiently well explained.

Point 1) My main concern, which I could not find the answer to, is why defectors are less likely to join a costly institute? Is this a finding, or an assumption of the model? I could not find such an assumption, but struggle to realize why defectors do not join. If there is a sufficiently large benefit to being in the costly institute, then all individuals should be willing to pay it, regardless of type. Unless defectors somehow have to pay more or receive less benefit but this does not seem to be the case. As this seems crucial to understanding the validity and importance of the results, I think this needs to be better explained.

Point 2) I'm also unsure on the value of the structured population analyses. I am not a modeller/theoretician, so maybe I am misunderstanding something, but how applicable is such a model to human scenarios? After all the paper is clearly mostly relevant to human cooperation (not many non humans punish and have social institutions). And the opening paragraph of the intro is clearly mostly about humans.

Point 3) I'm also surprised at the lack of consideration or mention given to the idea of relatedness. How can any theoretical study of the evolution of cooperation be complete with considering relatedness? My first instinct was that if the structured lattice population is more favourable to cooperation than the freely mixed population, then this is probably due to a build up of relatedness. Is this the case or not?

Point 4) Again, I'm not a modeller, but would a Wright (1931) infinite-island model not be more relevant for humans than the lattice, which seems more relevant for non-mobile organisms perhaps?

Point 5) What is the overall result? Does the costly institute favor cooperation, in general?, or just in

that institute? how often does it favor cooperation, most of the time, or just sometimes?

Point 6) From what i can gather, the pattern was often one of a form of 'chasing', whereby the defectors effectively chase, through evolutionary time, the cooperators, from one institution to the other, so they can exploit them. The cooperators in turn evolve to run away, in evolutionary time (by mutating game preference). This seems reminiscent of host-parasite models of cyclical co-evolution (negative frequency dependent selection). This seems potentially interesting.

If I'm uncertain it's largely because the paper does not help the reader much, the intro is largely just a repeat of the abstract, both saying the results rather than setting up the question, and then the result are frankly, too long. Too many figures, too much methods, and lack of signposting. The section on institutional quality comes out of nowhere and seems to be a detour from main narrative. The end of the results sound potentially important or exciting, but unfortunately I'm not sure many readers will make it this far.

Point 7) Finally, the conclusion mentions many examples, largely human, that also seem to contain elements of partner choice, or 'group-admittance', but this is not a feature of your models or discussed.

So in my conclusion, my subjective opinion is that this is a promising piece of research of some interest, but in its current form it would struggle to hold my attention or give me confidence in a take home message. I'm sorry if I sound negative. In order to be constructive, I suggest 1) make the introduction do more work to set up the research question rather than just summarize results which repeats the abstract anyway, 2) make clearer the points regarding why defectors don't join costly institute, 3) edit and streamline the results to what is important, 4) reduce number of figures and also make them bigger, really hard to read! And figure 1 legend makes no mention of panels a:d, what are they for? 5) discuss, at least, the role of relatedness, and maybe partner choice.

Responses to review comments

I am thankful to all the reviewers for their constructive suggestions, which I have hopefully used to improve the manuscript. In the following please find my point by point responses to the review comments.

Response to Reviewer N. 1

Reviewer N. 1:

The paper studies a model where individuals follow cooperative or defective strategies in the classical public goods game. However, differing to the traditional models, individuals follow another strategy in parallel: either they join to the group with costly institution or the cost free group. That is, institution cost suffers both the cooperators and defectors. The author pointed out that this kind of heterogeneity in the game facilitates the cooperation, and can lead to the coexistence of cooperators and defectors in the well-mixed and the spatially structured model. The problem of maintaining of cooperation in public goods situation is one of the classical problem of game theory and evolutionary biology. There are numerous solutions to the problem. This present paper suggests an alternative, less studied way to maintain cooperation. While the problem is important and the studied mechanism is interesting, the implementation of the project is problematic for numerous points. Below I collected my major and the minor points.

Response:

I am thankful to the reviewer for his/her careful reading and assessment of the manuscript, for his/her constructive suggestions, and for finding the work of interest.

Reviewer N. 1:

Major points 1. The paper is not clear enough in several points. The introduction should give examples (see my point later) where the suggested mechanism can have crucial effect on maintaining group cooperation. The first paragraph is too compact and consequently it has huge logical jumps. There are several papers which focuses on the role of individual or environmental heterogeneity on the evolution of cooperation (see e.g. Kun ., Dieckmann, U. 2013 Resource heterogeneity can facilitate cooperation Nature Communications 4: 2453, Stilwell, P., O'Brien, S., Hesse, E., Lowe, C., Gardner, A. and Buckling, A. (2020), Resource heterogeneity and the evolution of public goods cooperation. Evolution Letters, 4: 155-163). I suggest to study these papers and references therein and cite them accordingly.

Response:

I have studied these papers and other related papers and have added new examples where the suggested mechanism can be of importance, as well as a discussion and reference to related works on public goods.

Reviewer N. 1:

2. The model description is not detailed enough. For example, it is not determined clearly in the main text how the alternative groups are formed. Similarly, it should be described in more detail how the agent based models work, both in the well-mixed and the spatial model.

Response:

I have added a sentence to the model description to address this question, and two paragraphs to the method section to describe agent based simulations. Also, I have added Matlab codes used to simulate the model in mixed and structured populations to the Supplementary Information.

Reviewer N. 1:

3. The studied model is only a variant of the model published by the author recently (Salahshour

PLOS Comp. Biol 2021). He investigates the situation when there is a constant cost for joining to one of the cooperating group in the present ms, while he studied the case when the maximal reward of the cooperation is different in different environments in the PLOS Comp. Biol. paper. The two models are very similar in other aspects. The author should analyse the similarities and differences between the two models both in the introduction and in the discussion.

Response:

As the reviewer notes, in the present manuscript, I study the effect of heterogeneity in participation cost. In addition, the present manuscript also considers a structured population, which was not studied in the PCB paper. Besides the question studied in the two manuscript being different, which leads to important differences resulting from introducing entrance costs to the model, such as the evolution of cooperation for low enhancement factors which is not the case in the PCB paper, the phenomenology of the two models (the model considered here and the one in PCB) are markedly different. Mainly: 1) A large cost removes the bistability of the system, an important feature of the PLoS CB model, and changes the phase diagram. 2) The model introduced here shows two qualitatively different periodic orbits, separated by a phase transition. In the revision, I have devoted a paragraph to the comparison of the two models to the discussion, and have mentioned the relation to the PCB paper in the introduction.

Reviewer N. 1:

4. The paper presents the dynamical behaviour of the system in detail (see Fig. 3-5) which was already investigated in the PLOS paper in a similar model. I think the main result is the coexistence of cooperators and defectors as a consequence of individuals can join to different groups. These dynamical issues are less important. The author should explain convincingly what is the reason of higher level of cooperation and coexistence of cooperators and defectors is typical in this system. I note here that if one considers a single environment with the same simple public goods game, then either cooperators ($r/g > c$) or the defectors ($r/g < c$) win the competition. The situation is characteristically different in the present model, but why? A deeper mathematical argumentation would be convincing.

Response:

I have thoroughly revised the Results Section and new figures are added to help providing a better intuition (Two new figures (Figs 1 and 2) are added. 2 panels are added to figures 3 and 5. Figure 4 is retained from the old version. The rest of figures are removed or moved to the supplementary information.). Discussions with mathematical arguments are added to clarify the reason behind the evolution of cooperation and benefit of cost.

The key to the answer to this question resides in group composition fluctuations. An intuition can be given as follows. In other words, in a group composed of n_C^1 costly cooperators and n_D^1 costly defectors (and $n_C^2 + n_D^2$ individuals who prefer the free resource), a defector's payoff in the costly institution is equal to $n_C^1 rc / (n_D^1 + n_C^1 + 1) - c_g$. For small enough value of n_C and/or large enough value of n_D this payoff can be negative, which is smaller than that of non-costly cooperators (which is equal to zero). On the other hand, a cooperator's payoff in the costly institution is equal to $(n_C^1 + 1)rc / (n_D^1 + n_C^1 + 1) - c_g - c$, which for large enough n_C^1 and/or small enough n_D^1 can be positive. This condition can occur with high probability when the frequency of those who prefer the costly institution is low and, which happen, for instance, in small enhancement factors where the majority of individuals are non-costly defectors (it is also necessary that $c_g > r - c$ so that in the best case that everybody cooperates in the costly institute and costly cooperators receive their maximum possible payoff, their payoff becomes larger than zero).

Reviewer N. 1:

5. My other general problem with the paper is, that I dont think this kind of mechanism is common

in biology and similarly I think it is rare in human population. The author mentions some examples where individuals can join alternative cooperative groups at the end of the discussion, but there is no cost of joining to the group or having an extra cost of being the member of one of the group in these examples. Further, we know that other mechanisms maintain the cooperation in the presented examples (private benefit of the cooperators, reputation, punishment, ostracism, partner preference, etc.).

Response:

I agree that both theoretical and empirical literature are poor on this kind of mechanism. However, arguably in realistic situations individuals often face different public goods at the same time, in the form of different activities or different groups. This fact, if true, implies it is an important task to study such understudied but prevalent heterogeneous cases. In the words of a recent study on microorganisms, "Although this body of work [i.e., past works on the subject] has greatly aided our understanding of the dynamics of public goods traits, the scenarios investigated in these studies remain, by and large, simplified approximations of what really goes on in natural microbial communities. In particular, most studies consider just one model trait at a time. Under natural conditions, however, bacteria have to juggle a portfolio of various different public goods, and hence, they could routinely find themselves simultaneously participating in multiple public good dilemmas on multiple fronts (Brown & Taylor, 2010; Mellbye & Schuster, 2014)."

In the revised version I have spent some time to bring examples where this kind of mechanism can be applicable in microbial populations. As I have argued in the revised version, cost heterogeneity in this context, which can originate from maintaining different public goods system (i.e. different iron uptake systems in the example I have raised in the manuscript), although to my knowledge is not studied so far, seem more like a realistic and natural condition but an exception. Given the recent and growing literature in this field on situations where different public goods coexist, I hope my study can help to achieve a theoretical understanding of this understudied phenomenon. As I have mentioned by some examples in the paper, the situation in many animal populations, and more prominently, in human societies seems to be similar in terms of the existence of various public goods to participate in. In the case of human populations, I have mentioned different mechanisms (such as a membership fee or the cost of instituting and maintaining a public goods system) which seem to be at work in human populations and can give rise to cost heterogeneity. I have also mentioned costly signals, used as a means of group admittance or social status, as another mechanism which can underlie a participation cost. In this framework, the cost paid to join a costly institution, can be thought of a costly signal (such as donation, generosity, or risk-taking behavior) which increases one's social admittance. Such a n interpretation seems to me to also provide a novel perspective on costly signals.

Reviewer N. 1:

1. The abstract should emphasize that this is a theoretical work. To say that individual have a choice between the groups is a bit misleading. Choosing a group is a heritable strategy.

Response:

I have clarified that this is a theoretical work. Wherever possible I have changed the wording. I understand using the word choice is a bit inconsistent with an evolutionary interpretation and might be an slight abuse of the word. However, one can argue choice is made in evolutionary time. In addition, using the word choice seems more consistent with an imitation dynamic which is used in spatial model.

Reviewer N. 1:

2. There are numerous typos in the text.

Response:

The text is carefully revised to remove typos and improve the language.

Reviewer N. 1:

3. Figure 1: Subfigures should be designated. c_g emerges here first without defining it before. You mentioned replicator dynamics. I suggest to refer to the method section here.

Response:

This figure is changed. However, symbols are defined and reference to Methods is inserted in the new caption.

Reviewer N. 1:

4. It is not clear what is the role of including mutation among the strategies. It would be interesting to study the dynamics when only selection is allowed (naturally all ρ -s should be bigger than zero initially).

Response:

This is done in the revised version. panel c in Fig. 3 and Supplementary figures SI.8 and SI. 9 are devoted to this purpose.

Reviewer N. 1:

5. It is not clear what does happen if actually only one cooperator individual is present in a group? I note here, that the story would be totally different if the reward is a nonlinear function of the number of cooperators.

Response:

The game is played even when one cooperator exist in the group. That is the cooperator receives a payoff $r - 1$ from the game.

Reviewer N. 1:

6. I would change mixed population expression to well-mixed.

Response:

This is applied in the revised manuscript.

Reviewer N. 1:

7. The experienced periodic behaviour can be simply the consequence of using discrete time dynamics in replicator dynamics. Using the differential equation version can easily eliminate the dynamical instability.

Response:

I have examined the continuous version of the replicator dynamics in the revised version, Figs. SI.5 and SI.6. Periodic fluctuations exist in the continuous replicator dynamics as well. In general, the amplitude and the region of the phase space where fluctuations are observed increases for small mutation rates, and with continuous dynamics, smaller mutation rates are necessary for periodic fluctuations.

Reviewer N. 1:

8. Results, first sentence: There is no analytical solution of the problem. The author solved the analytical model by numerical integration. For that reason the sentence The result of the simulations in finite populations in the same paragraph is not precise. The numerical simulations of the analytical model and the numerical simulations of the agent based model lead to similar result This will be more adequate.

Response:

This comment is applied by changing the wording.

Reviewer N. 1:

9. Results, second paragraph: The explanation of the result is absolutely not convincing. We need some mechanistic or at least intuitive explanation.

Response:

I have added mathematical and intuitive arguments in the revised version.

Reviewer N. 1:

10. Page 7. 4b should be 4d.

Response:

Due to the revisions this part is changed.

Reviewer N. 1:

11. Page 9. Fig. 8. It is not clear why a different update rule (exponential payoff) is used here.

Response:

The two model are qualitatively the same. I have used a different model because for large β it is less noisy and leads to more clear snapshots and videos. I have also presented a video for the probabilistic model where it can be seen the dynamics is qualitatively similar. Importantly also, an exponential update rule is widely studied in the literature and some mechanism for the evolution of cooperation do not hold for both update rules (such as in some models spatial selection under a birth-death update rule). I have extensively studied the exponential model in the supplementary information to show the results are valid for this update rule as well. This defends the robustness of the results and the mechanism introduced.

Reviewer N. 1:

12. Page 9. last par. It is rather trivial that costly cooperator dominates if $r_1 - c_g > r_2$.

Response:

This part is removed. However, its main purpose was to show the existence of a mutualistic relation between cooperators with different game preference (that costly cooperators although in small density play a constructive role by removing non-costly defectors).

Response to Reviewer N. 2**Reviewer N. 2:**

In "Evolution of cooperation in costly institutes" authors show that in a situation where individuals have a choice between a costly institute and a free institute to perform a collective action task, the existence of a participation cost promotes cooperation in the costly institute. This is counterintuitive, because despite paying for a participation cost, costly cooperators can outperform defectors, who predominantly join a free institute. What is more, cooperation is thereby promoted also in the free institute. Research also shows that in a structured population a mutualistic relation between cooperators with different institute preferences emerges, which also helps the evolution of cooperation.

How and why cooperation emergence in public goods dilemmas is an intensely investigated subject with obvious practical ramifications. Methods of computational biology, statistical physics, and network science have been applied successfully and with much effect recently to shed light on the problem from many different perspectives, and also to outline many different ways on how solutions in terms of enhanced cooperation could be obtained. In this sense, the study addresses a relevant setup, and it also delivers results that might be of interest to the readership of Communications Biology.

I have quite enjoyed reading this manuscript, but I also have a couple of comments that require revision.

Response:

I am thankful to the reviewer for his/her careful reading and assessment of the manuscript and for and for his/her constructive suggestions and for finding the manuscript of interest.

Reviewer N. 2:

[minor suggestions] 1) In the first place, I am missing an intuitive explanation of these very counterintuitive results. It is already unexpected that the existence of a participation cost promotes cooperation in the costly institute. But then also cooperation is promoted in the free institute. Such rather bold conclusions require the identification of the underlying mechanism, and I would ask the author to spell this out to me in response to this comment, and then also in the manuscript. Preferably already in the abstract as much as this is possible, but then also definitively during the results and in the

discussion.

Response:

I have added explanations to shed light on the mechanism and provide an intuitive and mathematical understanding of the phenomenology of the model in the result section and also in the introduction. In simple terms the mechanism can be summarized as follows. The group compositions and the number of individuals in each subgroup (costly and free institutions) are subject to fluctuations. This introduces a risk for costly defectors, as they may do not meet a high enough number of cooperators in their group (or with probability $(1 - \rho_C^1)^{g-1}$ may not meet cooperators at all) so as to achieve a high enough payoff to compensate for the entrance cost. In a similar vein, costly cooperators can enter groups with no costly defectors (or with a low number of costly defectors), in which case they receive a positive payoff. Thus, as long as the probability of formation of such mixed groups is small, costly defectors receive an, on average, negative payoff (caused by paying an entrance cost) but costly cooperators can reach a positive payoff (provided $r > 1 + c_g$). Thus, cooperation decreases the risk of receiving a negative payoff when the frequency of individuals who prefer the costly game is small (and thus the probability of formation of mixed groups of costly cooperators and costly defectors is small). This regime, that is small frequency of individuals who prefer the costly resource, occurs for small enhancement factors, as the majority of individuals are non-costly defectors in this region.

In other words, in a group composed of n_C^1 costly cooperators and n_D^1 costly defectors (and $n_C^2 + n_D^2$ individuals who prefer the free resource), a defector's payoff in the costly institution is equal to $n_C^1 rc / (n_D^1 + n_C^1 + 1) - c_g$. For small enough value of n_C and/or large enough value of n_D this payoff can be negative, which is smaller than that of non-costly cooperators (which is equal to zero). On the other hand, a cooperator's payoff in the costly institution is equal to $(n_C^1 + 1)rc / (n_D^1 + n_C^1 + 1) - c_g - c$, which for large enough n_C^1 and/or small enough n_D^1 can be positive. This condition can occur with high probability when the frequency of those who prefer the costly institution is low and, which happen, for instance, in small enhancement factors where the majority of individuals are non-costly defectors (it is also necessary that $c_g > r - c$ so that in the best case that everybody cooperates in the costly institute and costly cooperators receive their maximum possible payoff, their payoff becomes larger than zero).

Regarding the question that how a costly institution can help the evolution of cooperation in the free institution, as r increases, the frequency of individuals who prefer the costly institution increases. This decreases the effective size of the free institution. Consequently, the probability of formation of groups with no or a small number of non-costly defectors increases, and non-costly cooperators can reach a high payoff in such groups. This helps the evolution of cooperation in the free institution, for intermediate values of enhancement factors.

Reviewer N. 2:

2) In terms of closely related research related to cooperation in the public goods game and institutions, there are two research papers that have been overlooked, namely Phase diagrams for the spatial public goods game with pool punishment, Phys. Rev. E 83, 036101 (2011), and Sustainable institutionalized punishment requires elimination of second-order free-riders, Sci. Rep. 2, 344 (2012). In both cases institutions are referred to as pool strategies, but the concept behind the considerations is the same. Essentially a governing body that is above an individual strategy that exerts some form of punishment, reward, or similar.

Response:

I thank the reviewer to bringing my attention to these works. I have improved the reference to the literature in the revised version by referencing to these and other related works.

Reviewer N. 2:

3) It would also improve the paper if the figure captions would be made more self contained. In addition to what is shown for which parameter values, one could also consider a sentence or two

saying what is the main message of each figure.

Response:

I have revised the captions to make them more self-contained following this suggestion.

Response to Reviewer N. 3

Reviewer N. 3:

This paper analyzes a model where individuals have a choice of which institute (aka game) to join, one of which has a participant cost and one does not, and whether to cooperate or not (binary decision). The two traits mutate and evolve independently. Individuals inherit or copy the traits according to the relative payoff success of individuals in preceding generation, and the population can be well mixed or spatially structured on a network/lattice. An analysis with institutions that vary in quality is also included. The overall conclusions are that a costly institute can favour cooperation, under certain parameters (when profitability, or enhancement factor, not too high).

Overall, this paper is of quite broad interest, as it deals with the evolution of cooperation, and is particularly orientated towards human type problems and solutions to cooperation. I also found the model description well written. However, I unfortunately find the whole paper quite difficult to digest, primarily the results, and some key points do not seem to be sufficiently well explained.

Response:

I am thankful to the reviewer for his/her careful reading and assessment of the manuscript, and for his/her constructive suggestions.

Reviewer N. 3:

Point 1) My main concern, which i could not find the answer to, is why defectors are less likely to join a costly institute? Is this a finding, or an assumption of the model? I could not find such an assumption, but struggle to realize why defectors do not join. If there is a sufficiently large benefit to being in the costly institute, then all individuals should be willing to pay it, regardless of type. Unless defectors somehow have to pay more or receive less benefit but this does not seem to be the case. As this seems crucial to understanding the validity and importance of the results, i think this needs to be better explained.

Response:

That defectors are less likely to join the costly institute is not an assumption of the model, but a result. The reason is that with some probability $((1 - \rho_C)^{g-1})$ defectors join a group in which no costly cooperator (or a too small number of costly cooperators) exists, in which case they receive a negative payoff of $-c_g$. (Even if they join a group with costly cooperators, depending on the number of costly cooperators and costly defectors in the group, r and c_g , their payoff can still be negative which is smaller than the minimum possible payoff of non-costly defectors. More precisely, in a group composed of n_C^1 costly cooperators, n_D^1 costly defectors, and $n_C^2 + n_D^2$ non-costly cooperator and non-costly defectors, a costly defector's return from the game, $n_C^1 r c / (n_D^1 + n_C^1)$ can be smaller than the participation cost c_g when n_D^1 is too high and/or n_C^1 is too low). As long as the density of costly cooperators is small, this can happen with a high probability, making joining the costly institute a risky enterprise for defectors, and on average an inferior option to joining the free institute (the payoff from which can not lie below zero). It pays for the costly defectors to join the costly institute, as long as there are enough costly cooperators to exploit. Thus the density of costly cooperators determine a bound for the density of costly defectors. In other words, it worth to join a costly institute as long as the density of costly cooperators is high enough and that of costly defectors is low enough so as to ensure a non-negative payoff. Otherwise, defectors are better off joining the free institute where a non-negative payoff is guaranteed. This fact keeps the relative frequency of defectors in the costly institution smaller than defectors in the free institution.

Reviewer N. 3:

Point 2) I'm also unsure on the value of the structured population analyses. I am not a modeller/theoretician, so maybe i am misunderstanding something, but how applicable is such a model to human scenarios? After all the paper is clearly mostly relevant to human cooperation (not many non humans punish and have social institutions). And the opening paragraph of the intro is clearly mostly about humans.

Response:

Population structure is widely studied and is established as one of the most important mechanisms for the evolution of cooperation. However, importantly, population structure can lead to the evolution of cooperation only under an imitation dynamics. That is under a dynamics that each site imitates the strategy of one of its neighbors according to their payoffs. (equivalently, this dynamics can be thought of as a death-birth dynamics (first death then birth) in which first a site is chosen at random for death and then one of the neighboring sites is selected proportional to their payoff to reproduce an offspring to replace the focal site. The imitation or death-birth dynamics can be contrasted with a birth-death update rule (first birth then death, in which first an individual is selected from the population proportional to its payoff to reproduce an offspring and then the offspring replaces a randomly chosen neighbor of the parent.) which is known that does not give rise to cooperation. Thus, the value of population structure for the evolution of cooperation crucially depends on an imitation dynamics which is very relevant to social evolution. It can be argued, social evolution takes place by individuals imitating each-others, their neighbors, or more generally those with whom they have contact, exchange information, and from whom they can learn. A population structure, or a network of contacts seems essential in this process.

Besides, as I have emphasized in the revision, the mechanism introduced here can be relevant in non-human organisms, or instance, the social evolution in microorganisms, which I have discussed rather in depth. In such cases, the spatial structure seem important as well.

Reviewer N. 3:

Point 3) I'm also surprised at the lack of consideration or mention given to the idea of relatedness. How can any theoretical study of the evolution of cooperation be complete with considering relatedness? My first instinct was that if the structured lattice population is more favourable to cooperation than the freely mixed population, then this is probably due to a build up of relatedness. Is this the case or not?

Response:

Relatedness (or kin selection) is a well-known mechanism for the evolution of cooperation. While in many cases, such as high level of cooperation in eusocial species, relatedness is probably the main mechanism at work to promote cooperation. In many other cases, such as human cooperation, or cooperation in many other animal populations, it is generally believed relatedness plays a limited or no role. Instead, in different contexts, different mechanism can be at work. For instance, direct reciprocity can be the main mechanism to promote cooperation in repeated interactions among unrelated individuals. Indirect reciprocity, punishment, and population structure, are yet other examples, each seem to be at work in different contexts. Indeed, in my manuscript, it is argued that heterogeneity in participation cost can be yet another mechanism to promote cooperation among unrelated individuals. Naturally, it can be expected this mechanism to be able to increase cooperation among related individuals as well, as relatedness can only ease the evolution of cooperation.

Reviewer N. 3:

Point 4) Again, I'm not a modeller, but would a Wright (1931) infinite-island model not be more relevant for humans than the lattice, which seems more relevant for non-mobile organisms perhaps?

Response:

It seems to me that depending on the context, different approaches could be more suitable for modeling

human interactions. Considering the high interest in evolutionary games on spatial structure it seemed to me studying the model on a spatial structure can be of more interest, and closer to a baseline case. In addition lattices have the virtue of simplicity which helps to assure the phenomenology observed is not caused by complicated population structures and are highly reproducible. While studying the effect of other population structures, such as heterogeneous social networks or an infinite island model, can be an interesting subject, using lattice seemed to me to be a reasonable choice for a first step to understand the phenomenology of the model.

Reviewer N. 3:

Point 5) What is the overall result? Does the costly institute favor cooperation, in general?, or just in that institute? how often does it favor cooperation, most of the time, or just sometimes?

Response:

A participation cost always suppress defection in the costly institute, in the sense that the average number of defectors in the costly institutes remains smaller than that in the free institute and the ratio of cooperators to defectors remains larger in the costly institute than its alternative free institute. A costly institute promotes cooperation for small enhancement factors, in the sense that in this region, cooperation does not evolve without a cost (in other words cooperation only evolves in the costly institute). For medium enhancement factors a costly institute also favors cooperation in its alternative free institution. For large enhancement factors, although the number of defectors and the ratio of cooperators to defectors still remain smaller in the costly institute, but fewer individuals prefer the costly institute. This increases the effective size of the alternative free institute and hinders cooperation in the free institute. In the revision, I have spelled out these results and have discussed different regimes and their differences.

Reviewer N. 3:

Point 6) From what i can gather, the pattern was often one of a form of 'chasing', whereby the defectors effectively chase, through evolutionary time, the cooperators, from one institution to the other, so they can exploit them. The cooperators in turn evolve to run away, in evolutionary time (by mutating game preference). This seems reminiscent of host-parasite models of cyclical co-evolution (negative frequency dependent selection). This seems potentially interesting.

If I'm uncertain it's largely because the paper does not help the reader much, the intro is largely just a repeat of the abstract, both saying the results rather than setting up the question, and then the result are frankly, too long. Too many figures, too much methods, and lack of signposting. The section on institutional quality comes out of nowhere and seems to be a detour from main narrative. The end of the results sound potentially important or exciting, but unfortunately I'm not sure many readers will make it this far.

Response:

I am thankful to the reviewer for this interesting observation. In the revision I have paid more attention to this point and have discussed two different regimes, red queen dynamics, resulted from negative frequency dependent selection, and black queen dynamics, resulted from a mutualism between cooperators with different game preferences.

Reviewer N. 3:

Point 7) Finally, the conclusion mentions many examples, largely human, that also seem to contain elements of partner choice, or 'group-admittance', but this is not a feature of your models or discussed.

Response:

I have mentioned group admittance as a suggestion for a mechanism that can cause a sort of participation cost. This can be construed as more rigorous in cases where group admittance requires a sort of costly donation, which seems to hold in some cases for example through providing costly signals for social status and group admittance. Based on this fruitful comment, I have clarified the discussion in

this regard and made a connection with costly signals of social admittance as a mechanism that can underlie a participation cost.

Reviewer N. 3:

So in my conclusion, my subjective opinion is that this is a promising piece of research of some interest, but in its current form it would struggle to hold my attention or give me confidence in a take home message. I'm sorry if I sound negative. In order to be constructive, I suggest 1) make the introduction do more work to set up the research question rather than just summarize results which repeats the abstract anyway, 2) make clearer the points regarding why defectors don't join costly institute, 3) edit and streamline the results to what is important, 4) reduce number of figures and also make them bigger, really hard to read! And figure 1 legend makes no mention of panels a:d, what are they for? 5) discuss, at least, the role of relatedness, and maybe partner choice.

Response:

I thank the reviewer once again for his/her constructive suggestions. In the revision, I have tried to apply these suggestions. Particularly, 1) the revised introduction both sets-up the question by referring to empirical examples (for instance in bacterial populations and humans) and the literature on heterogeneity. To ease the reading, I have also tried to spell out the results in the introduction and give an intuition for the results from the outset. 2) In the result section, using simple mathematical and intuitive arguments I have tried to explain why defectors do not join the costly institute. 3) The paper is thoroughly revised and I hope the result section, as well as other sections, now have a better organization. 4) The figures are also revised. Two new figures are added. and a panel substituted to figures 3 and 5. Figure 4 is retained. The rest of figures are moved to Supplementary information (in cases they did not appear repetitive in the SI) the revised version now has fewer hopefully more effective figures. 5) In the revision, the role of partner choice (group admittance) is briefly discussed in the discussion and the role of relatedness in the introduction.

REVIEWERS' COMMENTS:

Reviewer #1 (Remarks to the Author):

The author considered most of the issues carefully, this I think the paper improved significantly. However, I am still not convinced that the model has too much reality itself. The author argues that different iron uptake systems in bacteria or membership fee in human societies can be good examples for such mechanism. The model assumes that individuals can choose between two options: costly and cost free institutions, which helps the separation of cooperators and defectors. I think these alternative options within the same game framework is missing in the examples he mentioned.

Reviewer #2 (Remarks to the Author):

The author has revised his manuscript comprehensively and with love to detail. I warmly recommend acceptance in present form.

Responses to review comments

Response to Reviewer N. 1

Reviewer N. 1:

The author considered most of the issues carefully, this I think the paper improved significantly. However, I am still not convinced that the model has too much reality itself. The author argues that different iron uptake systems in bacteria or membership fee in human societies can be good examples for such mechanism. The model assumes that individuals can choose between two options: costly and cost free institutions, which helps the separation of cooperators and defectors. I think these alternative options within the same game framework is missing in the examples he mentioned.

Response:

I would like to thank the reviewer once again for his/her fruitful comments, which I have used to improve my paper. To address the generalizability and applicability of the model, in the second revision I have noted that a similar phenomenology is at work when instead of a free and a costly institute, two costly institutes with heterogeneous costs coexist. This is the case, because, when both institutions are costly, it is possible to absorb the cost of the cheaper institution in the base fitness. Thus the model becomes equivalent to a case where a costly and a free institution coexist but with a shifted base fitness. I have mentioned this in a sentence in the introduction, discussion, and a paragraph in the result section. I have also added a figure and a section to the Supplementary information to further document this point. I hope this increases the applicability of the model, and further improves the paper, as such a context seems a relevant situation empirically.

Response to Reviewer N. 2

Reviewer N. 2:

The author has revised his manuscript comprehensively and with love to detail. I warmly recommend acceptance in present form.

Response:

I thank the reviewer once again for his/her fruitful comments which I have used to improve the manuscript, and for finding the manuscript acceptable.